# How Big Is the Yeast Prion Universe?

**DOI:** 10.3390/ijms241411651

**Published:** 2023-07-19

**Authors:** Galina A. Zhouravleva, Stanislav A. Bondarev, Nina P. Trubitsina

**Affiliations:** 1Department of Genetics and Biotechnology, St. Petersburg State University, 199034 St. Petersburg, Russia; s.bondarev@spbu.ru (S.A.B.); n.trubitsina@spbu.ru (N.P.T.); 2Laboratory of Amyloid Biology, St. Petersburg State University, 199034 St. Petersburg, Russia

**Keywords:** yeast, prion, amyloid, protein aggregation, glutamine, asparagine, bioinformatics, proteome screenings, evolution

## Abstract

The number of yeast prions and prion-like proteins described since 1994 has grown from two to nearly twenty. If in the early years most scientists working with the classic mammalian prion, PrP^Sc^, were skeptical about the possibility of using the term prion to refer to yeast cytoplasmic elements with unusual properties, it is now clear that prion-like phenomena are widespread and that yeast can serve as a convenient model for studying them. Here we give a brief overview of the yeast prions discovered so far and focus our attention to the various approaches used to identify them. The prospects for the discovery of new yeast prions are also discussed.

## 1. Introduction

The term “prion” (from proteinaceous infectious particles) was proposed by Stanley Prusiner [1] to explain the unusual transmission properties of the pathogen observed in some neurodegenerative diseases in mammals (reviewed in [2]). According to Stanley Prusiner, “prions are proteins that adopt alternative conformations, which are self-propagating and found in organisms ranging from yeast to humans” [3]. The “prion diseases” are caused by conversion of the cellular protein, PrP^C^ into its abnormal isoform PrP^Sc^ (from SCrapie) and included human Creutzfeldt-Jakob disease, fatal familial insomnia, bovine spongiform encephalopathy (BSE) (or mad cow disease), sheep scrapie disease, and some other rare disorders (reviewed in [2,4]). Animal prion diseases have been diagnosed worldwide in various species, including mink, deer and camels, raising the question of the possibility of cross-species transmission as well as the potential risk to humans. However, the possibility of transmission of BSE from animals to humans has not been proved (reviewed in [5,6]).

Since 1982 when prion diseases were classified only as disorders caused by PrP^Sc^ aggregation, other neurodegenerative diseases such as Alzheimer’s or Parkinson’s diseases are often referred as prion diseases [3,7,8,9]. However, this idea remains controversial. Prions similar in their properties to PrP^Sc^ were found in various species, but their discovery in yeast has been inspired by Reed Wikner’s visionary description of yeast extrachromosomal elements [*URE3*] and [*PSI${}^{+}$*] as prions [10]. Since then, many new prions and prion-like proteins have been identified in yeast. Several recent reviews have focused on yeast prions [11,12,13,14,15,16,17,18,19]. Here, we provide a brief overview of the yeast prions found so far and focus our attention to the various approaches used to identify them. We also discuss the prospects for the discovery of new yeast prions.

## 2. Yeast Prions

Reed Wikner proposed several genetic criteria to classify a cytoplasmic determinant as a prion (Figure 1A): (1) reversible curability; (2) increased frequency of de novo prion generation in response to overproduction of the host maintenance protein; (3) requirement of the maintenance gene for prion propagation; (4) similarity between phenotypes produced by mutations in the maintenance gene and the prion-associated phenotype [10,20]. Although in subsequent years the number of these criteria could change, their general meaning remained the same, and all newly described prions were tested according to these properties.

Various approaches have been used to describe yeast prions, the main of which we have grouped as follows: (1) prions, discovered by their phenotype; (2) prions, isolated from systematic screening for the identification of proteins with glutamine/asparagine Q/N-rich sequences; (3) prions, allowing the appearance of other prions; (4) prions identified by combination of different approaches; (5) non-amyloid prions discovered as a beneficial hereditary factor (Figure 1).

From the biochemical point of view prion conversion in many cases is based on conformational changes of corresponding proteins (for a review, see [16,21,22]). There is a single example when the protein digestion (prion [β]) is required for prion appearance [23]. Conformational changes associated with prionization often lead to the formation of amyloid aggregates, however most recent examples demonstrated that the prion aggregates may be non-amyloid ([24], for review, see [16]).

### 2.1. Prions, Discovered by Their Phenotype

The discovery of yeast prions is connected with the identification of inheritable cytoplasmic factors with unusual properties (Figure 1A). [*PSI${}^{+}$*] was first described by Brian Cox at 1965 as a cytoplasmic allosuppressor [25] (for a review, see [26]). For a long time, the molecular nature of this factor remained enigmatic, until in 1994 it was shown that the [*PSI${}^{+}$*] factor is dependent on the *SUP35* gene [27]. Shortly thereafter, R. Wikner suggested that the Sup35 protein could be a yeast prion (see review [28]).

Sup35 (or eRF3) is a translation termination (or release) factor [29,30]. In yeast *Saccharomyces cerevisiae* the *SUP35* gene is essential, as well the *SUP45* gene, encoding the translation termination factor eRF1. While eRF1 is thought to play a major role in termination [31], eRF3 is a GTPase that binds to eRF1 and stimulates its release activity (reviewed in [32,33]). Yeast Sup35 is a 685 amino acids protein that consists of three regions with different functions. The N-proximal region (Sup35N) is not important for viability and termination, but it is required for [*PSI${}^{+}$*] induction and propagation, thus it is considered a PrD (a prion-determining domain) (Table 1, Figure 2) ([27,34,35], reviewed in [26,36]).

Sup35N contains two sub-domains: a Q/N-rich region (residues 1–40) and an oligopeptide repeat region (residues 41–97). Five full PQGGYQ(Q)QYN repeats and one partial repeat, PQGG, was found in this region after sequencing of the gene [37] (Figure 2), however more recent investigation revealed eight repeats (residues 28–118) [38]. Sup35M (middle) region is enriched with charged amino acids and is not required for viability and translation termination but is involved in the interaction with Hsp104 [39]. Hsp104 is a disaggregation-stimulating protein essential for propagation of [*PSI${}^{+}$*] [40] and most other yeast prions (reviewed in [21,22,41]). Many of the chaperones and different factors required for the propagation of yeast prions have been described and discussed in recent reviews [14,42,43], for these reasons they will not be discussed here. NM-domain forms reversible pH-dependent biomolecular condensates [44]. C-proximal region (Sup35C) has a significant similarity with the translation elongation factor eEF1A (Figure 2) [37,45,46] and is required for translation termination and cell viability [29,30,47]. Sup35C contains GTP-binding motifs and interaction sites with various proteins including eRF1 (see [33] for a recent review). [*PSI${}^{+}$*] remains the most studied yeast prion, widely used to characterize various aspects of prion biology, including the existence of different strains or variants (see [15] for recent review).

[*URE3*] was first described in 1971 by Francois Lacroute as the non-Mendelian mutation allowing cells to grow on a minimal medium supplemented with ureidosuccinic acid [48]. Later it was shown that [*URE3*] is the aggregated form of Ure2, a product of the chromosomal *URE2* gene (reviewed in [49]), working as a nitrogen catabolite repression transcriptional regulator [50]. Ure2 is 354 amino acids protein, and like Sup35, it also has the PrD region which is located within the N-proximal part of the protein (residues 1–80) and named “primary” PrD (Table 1, Figure 2) (reviewed in [28,51]). Ure2 N-terminal domain (Ure2N), is rich in N and Q, as well as Sup35N, although Ure2N contains more N than Q residues compared with Sup35N (reviewed in [36]). A secondary PrD within the C-proximal part of Ure2 was also found (Figure 2), it overlaps amino acid residues positions 221–227 and is activated by deletions of “prion-inhibiting” regions (residues 151–158 or residues 348–354), which are thought to stabilize the secondary PrD preventing its conversion to the prion form [52]. This important observation shows that regions not adjacent to the PrD may influence prion formation (reviewed in [53], see also [54], and references therein). Also the first 105 residues of Ure2 participate in the interaction with both subunits of the Cyc8-Tup1 corepressor [55] (Figure 2). Thus, the PrD (residues 1–80) of Ure2 overlaps with the site of Ure2 binding to the corepressor Cyc8-Tup1 (residues 1–105), and it is possible that the interaction of Cyc8-Tup1 with Ure2 can prevent the prionization of Ure2, and vice versa, the formation the [*URE3*] prion will interfere with the Cyc8-Tup1-Ure2 interaction. It is appropriate to note here that Cyc8 is able to form the [*OCT${}^{+}$*] prion and that Tup1 is a Q-rich protein (see below).

The [*ISP${}^{+}$*] prion (Inversion of Suppressor Phenotype) was detected by its ability to inhibit the suppressor phenotype of some *sup35* mutants [56]. The search for the protein that forms this prion led to the discovery of a suitable candidate, Sfp1 [57]. [*ISP${}^{+}$*] has a number of features which distinguish it from most “canonical” prions. Its propagation does not depend on the Hsp104 chaperone [56], which is required for propagation of other prions, and deletion of the *SFP1* gene results in a phenotype drastically different from the [*ISP${}^{+}$*] phenotype [57]. Further research has shown that the Isp^+^ phenotype may arise due to a change in the chromosome II copy number, and that overproduction of Sfp1 may induce such changes [58]. The history of [*ISP${}^{+}$*] clearly demonstrates that conclusions about the amyloid nature of a particular prion, based on genetic analysis, but not supported by significant biochemical or structural data, may not be convincing enough.

[*GAR${}^{+}$*] (from “resistant to Glucose-Associated Repression”) was identified as a prion which turns off the glucose repression [59]. This factor allows cells to utilize the poor carbon sources in the presence of rich ones. In particular [*GAR${}^{+}$*] cells are able to grow on medium with glucosamine and glycerol as a sole carbon source (we will further refer to such phenotype as [*GAR${}^{+}$*] phenotype). Also this prion decreases expression of *HXT3*. Noteworthy that [*GAR${}^{+}$*] propagation is unrelated to Hsp104 but depends on Hsp70s activity [59]. Search of prion protein revealed that this factor comprises two components and this is its unique feature (Table 1). Overproduction of Std1, a protein involved in the regulation of glucose metabolism, increased the appearance of [*GAR${}^{+}$*] colonies, however *STD1* is not essential for the prion propagation. At the same time plasma membrane ATPase Pma1 is bound with the Std1 in [*GAR${}^{+}$*] but not in [*gar${}^{-}$*] strain. The mild overexpression of *PMA1* led to an increase in appearance of [*GAR${}^{+}$*], and the detailed mutation analysis supported the role of this gene in the prion maintenance. Thus, it was proposed that these two proteins are required for the prion propagation, indeed the strain Δ*std1* and *PMA1*Δ*40N* (N-terminus of Pma1 is predicted as unstructured) is unable to propagate [*GAR${}^{+}$*] [59]. [*GAR${}^{+}$*] cells can appear spontaneously [59], but also the appearance of cells with [*GAR${}^{+}$*] phenotype is elevated by the bacteria (*Staphylococcus hominis*, *Listeria innocua*, *Sinorhizobium meliloti*, *Bacillus megaterium* and others) metabolites, especially lactic acid [60,61]. Cells with [*GAR${}^{+}$*] phenotypes can be isolated in different yeast strains and even species [60,62]. [*GAR${}^{+}$*] may affect cell morphology and size, as well as distribution of the Hxt3 and Pma1 in cells. Also the prion changes the compositions of phospholipids and fatty acids and reduces oxygen uptake [63]. The [*GAR${}^{+}$*] prion allows cells to grow well in more variable environmental conditions by switching the metabolic strategy from “specialist” to “generalist” in utilization of different carbon sources [60]. However the [*GAR${}^{+}$*] mediated effect on fermentation efficiency was not supported by further investigations and thus may be restricted by specific genetic background of yeast strains ([62,64], reviewed in [65]).

The [β] prion was the first one whose propagation is not based on the conformational change of the protein but on its digestion. The activation of the CpY hydrolases in yeasts depends on activity of PrA or PrB. In turn the activation of PrB (encoded by *PRB1*) requires mature PrA (*PEP4*) [66]. Thus the CpY activity usually was used to detect *pep4*Δ. Noteworthy is that the activity of PrB and CpY was preserved after at least 20 generations in cells with *PEP4* deletion [67]. However the indefinite CpY activity was found in the meiotic progeny of *pep4*Δ/*PEP4* diploids and then it was shown that the active PrB can support the activation of CpY. Remarkably, the CpY activity can be restored in *pep4*Δ mutants by cytoplasm transfer from PrB^+^ donor. Thus the existence of a cytoplasmic factor, called [β], was proposed. Further investigations revealed that the *PRB1* is required for the maintenance of this factor and it satisfies all prion criteria (Table 1). The molecular mechanism underlying the [β] phenotype is an in trans activation of pro-PrB by already cleaved PrB [23].

### 2.2. First Prions, Isolated from Systematic Screening for the Identification of Proteins with Q/N-Rich Sequences

A search of genomic databases for proteins with high amounts of Q/N, similar to those in PrD of Sup35 and Ure2, revealed New1 and Rnq1 as potential prion proteins [68,69]. In both cases fusions with heterologous protein (Sup35) were used to demonstrate prion function (Figure 1B). New1 contains the PrD in its N-terminal part (residues 1–153) (Table 1). This PrD fused to the GFP or MC-domains of Sup35 was able to induce aggregation of the reporter protein [68,70]. Also PrD of New1 (residues 1–130) stimulated aggregation of Sup35NM [71]. Since no phenotypic changes are associated with New1 overexpression or inactivation, it is difficult to test the possibility of New1 to form a prion on its own. For this reason, [*NU${}^{+}$*] is a prion-like protein formed by the PrD of New1 (residues 1–153) fused to the MC-domains of Sup35 [68]. The New1 induces fragmentation of Sup35NM fibers in vitro and its overproduction changes the morphology of Sup35NM aggregates in [*PSI${}^{+}$*] cells [71]. Interestingly, the C-terminal domain of New1 is homologous to translation elongation factors and participates in the translation termination or ribosome recycling [72].

[*RNQ${}^{+}$*] was identified after a BLAST search, followed by expression analysis and subsequent fusion of the identified proteins to GFP. Along with Rnq1, two more proteins (YBR016w and Hrp1), were described that showed GFP aggregation, but their properties were not further studied [69]. Rnq1 (Rich in N and Q) contains the PrD in its C-terminal Q/N-rich region (residues 153–405) [69] (Table 1, Figure 2). However, it appears that the N-terminal portion of Rnq1 is also required for prionization and that the C-terminal PrD is more complex in its organization than the Sup35 PrD (reviewed in [73]). Aggregation of Rnq1 leads to formation of prion [*PIN${}^{+}$*] [74].

### 2.3. Prions, Allowing the Appearance of Other Prions

The Prion [*PIN${}^{+}$*] (from [*PSI${}^{+}$*] INducibility) is required for efficient [*PSI${}^{+}$*] induction by overproduced Sup35, although some (but not all) constructs containing Sup35N may overcome the [*PIN${}^{+}$*] requirement [74,75,76] (Table 1). In the screening for proteins that, when overexpressed, can contribute to the formation of [*PSI${}^{+}$*], [*PIN${}^{+}$*] was shown to be associated with an aggregated form of the Rnq1 protein (Figure 1C) [74]. This work also identified several other proteins that can function as [*PIN${}^{+}$*] and promote the formation of [*PSI${}^{+}$*], among them New1 and Ure2, as well as proteins later described as prions: Swi1 [*SWI${}^{+}$*], Cyc8 [*OCT${}^{+}$*], Pin3 [*LSB${}^{+}$*].

The Prion properties of [*SWI${}^{+}$*] were described in 2008 [77]. Swi1 regulates transcription of many genes being a subunit of the SWI/SNF chromatin remodeling complex, and thus, in its soluble form, Swi1 is present in the nucleus. Aggregation of Swi1 leads to formation of [*SWI${}^{+}$*] prion localized in the cytoplasm and manifests in reduced growth on non-glucose carbon sources (reviewed in [78]). Yeast Swi1 is a 1314 amino acid protein that consists of three regions with different functions (Table 1, Figure 2). The N-proximal region (N, residues 1–323), is sufficient for [*SWI${}^{+}$*] formation and propagation, the middle glutamine-rich region (Q, residues 342–524), modulates the activities of the N-terminal and C-terminal domains, and the C-terminal region (C, residues 525–1314) is required for chromatin remodeling function [79] (Figure 2). The Swi1 PrD can be truncated to an N-terminal fragment, residues 1–38. It was shown that the first 38 amino acids of Swi1 are sufficient for propagation of [*SWI${}^{+}$*], interaction with [*SWI${}^{+}$*] aggregates, the ability to aggregate and maintain folding of [*SWI${}^{+}$*], and also for de novo prion formation (see [80] and references therein). Interaction between [*SWI${}^{+}$*] and [*RNQ${}^{+}$*] prions leads to the formation of a new prion, [*NSI${}^{+}$*] [81].

The Prion [*OCT${}^{+}$*] is formed by the transcriptional co-repressor Cyc8 (or Ssn6) which, together with another protein, Tup1, acts as a global transcription regulator [82]. Cyc8 is a 966 amino acid protein consisting of an N-terminal part essential for its function and a dispensable C-part (Table 1, Figure 2). The PrD of Cyc8 is located in its C-terminal part and includes residues 465–966 [82], or more precisely, residues 443–672 [83]. Near its N-terminus, the Cyc8 protein contains 10 tandem copies of the TPR (tetratricopeptide repeat) motif (TRP-like domain), which is required for the interaction of Cyc8 with Tup1 and other repressor proteins (see [84] and references therein). Cyc8 is also characterized by the presence of several Q-rich regions. The first is N-terminal, located before the TRP-like domain and contains 16 glutamine residues (Qx16), the second is situated in the central region (residues 493–598). Deletion of the Qx16 tail of Ssn6 (Cyc8) results in its self-oligomerization which prevents the Ssn6-Tup1 interaction and its transcriptional repression activity [84]. Connection of this aggregation with prionization of Cyc8 and appearance of prion [*OCT${}^{+}$*] was not studied. Truncated from C-term (after PrD) variants of Cyc8 were also able to aggregate but not to convert the full-length Cyc8 into the aggregated form [85]. Interestingly, Tup1 is also a Q-rich protein (see [86]).

Pin3 has been identified as a protein that can promote the formation of [*PSI${}^{+}$*] when overproduced [74] and later the corresponding prion was named [*LSB${}^{+}$*] [87]. Pin3 is a short protein (215 residues) with potential PrD located in its C-terminal part (residues 113–183) [83] (Table 1, Figure 2). According to *Saccharomyces* Genome Database, Pin3 is a short-lived protein whose levels increase in response to thermal stress, it interacts with Las17 via an N-terminal SH3 domain, and cooperatively inhibits the nucleation of actin filaments. The [*LSB${}^{+}$*] prion was characterized by low mitotic stability, but had all the typical prion properties [87].

The [*MOD${}^{+}$*] prion has also been isolated in the search for prions, allowing the appearance of other prions. However, in this case, the authors used not the induction of [*PSI${}^{+}$*] by overproduced Sup35, but the induction of [*Q${}^{+}$*] factors by overproduction of the Sup35 chimeric protein containing 62 glutamine repeats instead of the first 40 residues at the N-terminus of Sup35 [88] (Figure 1C). Mod5 protein, responsible for the formation of [*MOD${}^{+}$*], is a tRNA modification enzyme, tRNA isopentenyltransferase. The 428 residues of Mod5 sequence does not contain Q/N-rich domains; however, Mod5 has an amyloid core mapped in the region of residues 194–215 (Table 1, Figure 2). [*MOD${}^{+}$*] cells are resistant to several antifungal drugs [89]. Mod5 binds directly to both substrate and non-substrate RNAs, and binding of Mod5 to tRNA-like molecules facilitates Mod5 aggregation [90].

### 2.4. Prions Identified by Combination of Different Approaches

The [*MOT3${}^{+}$*] prion was identified by a combination of bioinformatic screening followed by various molecular biological assays (Figure 1D) [83]. Mot3 (from MOdulator of Transcription) is a global transcription factor that can act as a repressor or activator and is involved in the regulation of various processes in the yeast cell [91]. Mot3 is a 490 amino acid protein consisting of several Q, N, and P stretches along with two C2H2 zinc fingers at its C-terminus; this structure is typical of other transcriptional repressors [91] (Table 1, Figure 2). The repression domain of Mot3 was placed at residues 231–347 [55]. Potential PrD of Mot3 is located in its N-terminal part (residues 1–259) [83,92]. Thus N-part resembles the non-structured domain of transcription factors and also participates in prionization, while C-part is involved in DNA-binding. The [*MOT3${}^{+}$*] prion is induced by ethanol and eliminated by hypoxia and regulates the expression of flocculin *FLO11* [92]. Two other yeast prion proteins, Swi1 and Cyc8, are also regulators of *FLO11* gene expression and, hence, the multicellular properties of yeast cells (reviewed in [78]). Interestingly, Mot3 binds to Cyc8 in vitro [55], thus providing possible interactions between [*MOT3${}^{+}$*] and [*OCT${}^{+}$*] prions.

Several other potential prions have also been predicted, including [*NUP100${}^{+}$*] and [*GLN3*] [83]. However, the prion formation by GLFG-nucleoporins or Gln3 have been shown only under conditions of overproduction of the corresponding protein and/or its prion-forming domain [93,94]. Thus, the ability of these proteins to form prions by themselves remains unclear.

### 2.5. Non-Amyloid Prions Discovered as a Beneficial Inheritable Factor

A lot of amyloid-based yeast prions were discovered at the end of 20th century and at the beginning of the new one and the wide-scale screening (Figure 1D) revealed numerous additional candidates [83]. The following search of prions [24] had demonstrated that transient overproduction of many intrinsically disordered proteins in yeast can lead to the appearance of heritable beneficial traits (Figure 1E). This screening tried to overcome the limitations of previous ones: (i) focus only on Q/N-rich proteins, which are able to form amyloid (ii) obligatory implication of Hsp104 in prion propagation. Yeast cells overexpressing one of known ORF (approximately 5300 constructruction were analyzed) were cultivated in ten different stress conditions. Such analysis revealed that hundreds of proteins ameliorate cell growth. Moreover in 80 cases the ancestors with normal protein expression obtained new phenotypic traits, 46 of them were stable and passed through at least 100 generations. The non-Mendelian inheritance and transmission by cytoduction were investigated and proved for 18 potential prions (there is only one known prion, [*MOT3${}^{+}$*], among them). All of them can be cured by alterations of Hsp70 and Hsp90 activities. Finally it was shown that corresponding proteins do not form amyloids, but aggregate. Many of identified prions provided new functions to cells compared to the deletion of corresponding genes. For example, overproduction of transcription factor Azf1 led to the inheritable resistance to radicicol. Corresponding prion was called [*AZF1${}^{+}$*] [24]. Interestingly, the polyN domain of Azf1 is not responsible for its prionization [95].

Proteins revealed in this screening as prions were enriched by intrinsically disordered regions. This property is conservative in evolution, thus authors conclude that prion phenomena may be much more widespread than is assumed now [24]. Also noteworthy that discovered factors were then found in different yeast strains including natural isolates [96,97]. Below we describe several examples studied in more detail.

The [*SMAUG${}^{+}$*] prion is induced by transient overproduction of Vts1 protein (its ortholog in *Drosophila melanogaster* is called Smaug) [24,98]. Prionization of Vts1 led to its hyperactivation and more efficient degradation of mRNA with Smaug recognition element (SRE) [96,98]. This protein forms condensates in vitro, which are able to bind mRNAs with SRE. The complexes are non amyloid but infectious and can be easily disassembled [98]. [*SMAUG${}^{+}$*] is advantageous when the concentration of glucose in the environment is low [98] and favors the proliferation of yeast cells compared to sporulation [96]. Transcriptome-wide analysis revealed changes in expression of more than 200 genes [96] in the prion strain and repression of *MUM2* [96]. The further interest in this gene is explained by several reasons: it is overexpressed in *vts1*Δ strain, its mRNA contains SRE element, and it is linked to the sporulation (*mum2*Δ cells demonstrate sporulation defects). Target verification confirmed the *MUM2* repression in [*SMAUG${}^{+}$*] cells during the first 14 hours after sporulation induction. Further analysis revealed that strong expression of *MUM2* restored the sporulation in the [*SMAUG${}^{+}$*] cells. The normal SRE is required for the *MUM2* downregulation in such strains. Thus the [*SMAUG${}^{+}$*] prion promotes cells proliferation and sporulation delay via degradation of the *MUM2* mRNA [96].

The [*ESI${}^{+}$*] (Expressed Sub-telomeric Information) was found in the screening and provides zinc-resistance to the yeast cells. This factor appeared after Snt1 (component of the Set3C histone deacetylase complex) transient overproduction [24]. [*ESI${}^{+}$*] demonstrates non-Mendelian inheritance, is transmissible upon cytoduction, can be eliminated by perturbations in chaperone activities, and requires presence of Snt1 [99]. Snt1 acquires protease resistance in prion form, and this property can be transmitted on native protein in vitro. The protein can aggregate in vitro and these complexes can transform [*esi${}^{-}$*] to [*ESI${}^{+}$*] yeast cells. The appearance of the [*ESI${}^{+}$*] can be stimulated by the cell cycle arrest and subsequent phosphorylation of Snt1 [99]. The phenotypic manifestation of the prion is linked to changes in histone (H3 and H4) acetylation followed by an increase in transcription of approximately 1000 ORFs. Notably that such effect could not be explained by simple inactivation of Snt1 because expression changes in strains with loss of Set3C (including *snt1*Δ) function do not completely correspond to the [*ESI${}^{+}$*] strain. Many revealed overexpressed transcripts located in repressive, Hda1-affected sub-telomeric domains. ChIP-Seq analysis revealed that expression activation in such regions is accompanied by histone H4 acetylation and the recruiting of the RNA polymerase II, Hos2 (histone deacetylase) and Snt1. At the same time the Rap1 (repressor-activator site binding protein) binding to the subtelomeric regions is decreased in [*ESI${}^{+}$*] cells. The reduced amount of Rap1 phenocopy the prion phenotype in naive cells but does not affect the [*ESI${}^{+}$*] cells. Authors proposed that in [*ESI${}^{+}$*] cells phosphorylated Snt1 interfere with Rap1 and decrease the amount of the repressor in telomeric regions. This leads to the attraction of Set3C complex, H4 acetylation and recruitment of RNA pol II into the subtelomeric regions followed by increase in gene expression [99].

The [*BIG${}^{+}$*] (Better In Growth) factor is a prion form of Pus4 (pseudouridine synthase) [24,97]. This factor provided the resistance to the zinc sulfate and increase in proliferation rate but decreased chronological lifespan. [*big${}^{-}$*] and [*BIG${}^{+}$*] cells differ in their sizes in dense cultures. This factor is dominant and non-Mendelian. The appearance of the prion led to the change of Pus4 localization, so part of the protein can be found outside of nucleolus (normal localization) in the cytoplasm [97]. The [*BIG${}^{+}$*] phenotype differs from the manifestation of the *PUS4* deletion and Pus4 preserves its function in cells with this prion. Moreover it is likely that the Pus4 catalytic function may be increased for several mRNAs. This likely led to the subsequent changes in translation efficiency and corresponding phenotypes [97].

### 2.6. Interplay between Different Approaches

The approaches for the identification of yeast prions discussed in the previous section seems to be independent. However, indeed the discovery of first yeast prions and descriptions of their properties (Q/N-enrichment or amyloid properties of corresponding protein in prion isoform) encouraged followed analysis of proteins with the same characteristics. Discovery of [*PIN${}^{+}$*] and screening for corresponding proteins revealed several new prions. Finally the high-throughput screening for almost all yeast genes attempted to reveal all beneficial prions (Figure 3). Also it should be mentioned that prion investigation promotes the progress in the adjacent fields. Thus the identification of [*NSI${}^{+}$*] prion protein required the development of new proteome screening for proteins formed detergent-resistant aggregates [81,100].

## 3. Problems in Discovery of New Yeast Prions

Prions first became known in the context of human and animal neurodegenerative diseases [1,2]. Thus, they have attracted a lot of attention and, given their ambiguous nature, have been intensively studied. The discovery of prions in yeast gave rise to a huge amount of work on this subject. To date, the largest number of prions has been found in yeast *S. cerevisiae*. In a short time, about 10 of them were described, but after such a breakthrough, the discovery of new prions seemed to have stopped, despite the significant efforts that were made to detect them. The first discovered prions [*PSI${}^{+}$*] and [*URE3*] provided the basis for genetic and bioinformatic screens to find similar determinants in yeast. Above, we outlined the main directions for the identification of new prions: (1) search by phenotype; (2) systematic screening for Q/N-rich proteins; (3) search for [*PIN${}^{+}$*]-like determinants; (4) search using a combination of different approaches; (5) search for non-amyloid prions.

The first prions were found by chance, drawing attention to themselves due to unusual phenotypic manifestations. Nevertheless, more than two decades has passed from the moment they were first mentioned to being described as “prions”. This highlights the fact that genetic analysis is complex and sometimes resource and time consuming. At the same time, it can give controversial results, as was demonstrated above for the [*ISP${}^{+}$*] determinant [56,58]. However, without the use of genetic methods, we cannot accurately conclude whether a prion-like protein is a prion or not.

Extensive bioinformatic screening followed by a combination of various molecular genetic analysis identified 200 candidate proteins, of which only 18 passed all four selection criteria as potential prions, and for only one of them the prion properties of native protein were shown ([*MOT3${}^{+}$*]) [83]. More accurate and successful identification of prions requires the development of genetic screening techniques and, importantly, a deeper understanding of the biology of the object under study. Perhaps, as was previously done for the [*GAR${}^{+}$*] prion [59], we should turn to earlier genetic studies and try to comprehend previous knowledge using modern technologies. The description of the inheritance of a rare phenotype whose frequency of occurrence is higher than the occurrence of mutations may indicate the presence of an epigenetic switch, such as a prion.

The identification of the [*PIN${}^{+}$*] prion made it possible to search for factors capable of replacing [*PIN${}^{+}$*] in the formation of the [*PSI${}^{+}$*] prion and to identify a number of new potential prions [74] (see Section 2.3 above). Despite the success of this method, it has limitations for identifying new prions. The exact mechanism by which potential prion proteins can replace [*PIN${}^{+}$*] upon [*PSI${}^{+}$*] induction is unknown. The most reasonable is the cross-seeding model, according to which preformed oligomers of one protein accelerate the aggregation of another protein ([74], reviewed in [101]). The most recent data in this field were collected in the AmyloGraph database [102]. It is important to note that the most significant property of the PrD of many yeast prions is the predominance of Q/N residues in the PrD composition. It is the amino acid composition, and not the exact sequence, that determines the properties of the prion, since randomization of the amino acid order in the Ure2 or Sup35 prion domains did not adversely affect the prion formation or propagation [103,104,105]. However, the efficiency of prion formation by scrambled Ure2 and Sup35 differs from that of wild-type proteins (reviewed in [106]). This suggests that prionization can be influenced by primary sequence, including the presence of aromatic and hydrophobic amino acid residues [107,108], reviewed in [106]). Indeed, it has been shown that an artificial prion formed by Sup35 with a uniform poly-Q tract instead of PrD was efficiently fragmented in yeast when tyrosines (or some other residues) were interspersed in the poly-Q region [109]. These data suggested that prion domains could arise from polyQ (or polyN) sequences, followed by substitution of Q (or N) for fragmentation-promoting residues [110]. It was shown that in the Saccharomycetes clade, an evolutionary increase in the amount of Q/N-rich proteins is observed in comparison with all other clades of the kingdom of fungi [111]. Perhaps it is this feature that contributes to the abundance of prions in yeast. Thus it may be supposed that new QN-rich proteins with prion properties can be discovered in future. Previously we mentioned two non-investigated aggregation-prone candidates: YBR016w and Hrp1. The most recent screening [24] was aimed to identify almost all yeast proteins with prion properties. And we suppose that this goal was achieved in general, however the prions lacking obvious beneficial phenotype could be missed in this work.

## 4. Conclusions

So, what should be the right strategy for finding new yeast prions? Depends on what exactly we are looking for. The proteins that form prions in yeast, despite many common characteristics, have significant differences. Among them are the functions of the protein, the ratio of Q and N, their position relative to the protein sequence, or their complete absence, the amount of protein in the cell and its lifetime, etc. The term “prion”, like the term “amyloid” (see review by [112]), is probably undergoing its crisis. Attempts have already been made to understand and give a flexible classification of prions and prion-like proteins, dividing them into prions, quasi-prions, and prionoids [113]. However, so far our knowledge is not enough to put an end to this issue. Perhaps, by continuing to study prions in yeast, we will gain a deeper understanding of the complex history with the epigenetic regulation of the work of our genetic apparatus. Recently, no new prions have been discovered in yeast, probably because much more effort is required than was necessary at the initial stage of their study. We believe that the search for prions should be approached systematically, using a combination of different approaches. It is important to take into account not only the intramolecular context—the amino acid composition and environment of the PrD, but also the intracellular one—the potential interactions of the protein of interest in the cell.

## Figures and Tables

**Figure 1 ijms-24-11651-f001:**
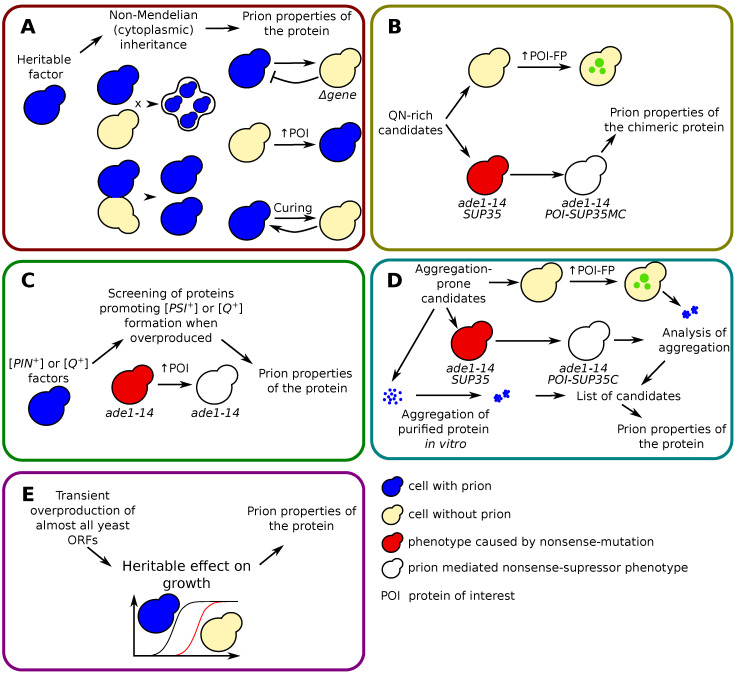
**Strategies for identification of prions or prion-like proteins.** (**A**) The description of non-Mendelian inheritable factors followed by identification of protein required for its propagation. (**B**) Investigation of prion properties of Q/N-rich proteins. Corresponding candidates are revealed with the bioinformatic analysis of yeast proteome. Aggregation of the protein fused with fluorescent protein is monitored with microscopy. The fusion of protein of interest with functional domain of Sup35 allows verify aggregation on the phenotypic level. Inactivation of the POI-Sup35MC into aggregates leads to the nonsense-suppressor phenotype which can be easily monitored by in cells with nonsense-mutations (for instance, *ade1-14*). (**C**) Screening for proteins, which can trigger formation of other prions. Appearance of [*PSI${}^{+}$*] and [*Q${}^{+}$*] factors, leading to the nonsense-suppressor phenotype, requires another prions. Identification of corresponding proteins was carried out with large scale screening when different genes were overexpressed. (**D**) Analysis of prion properties of aggregation-prone proteins by complex approach: analysis of aggregation of overproduced protein fragments fused with GFP, appearance of nonsense-suppressor phenotype for the fusion of the protein and functional domain of Sup35, formation of amyloid aggregates in vitro. (**E**) Systematic screening for inheritable factors induced by transient overproduction of yeast ORFs. Each approach is illustrated with examples below in the text.

**Figure 2 ijms-24-11651-f002:**
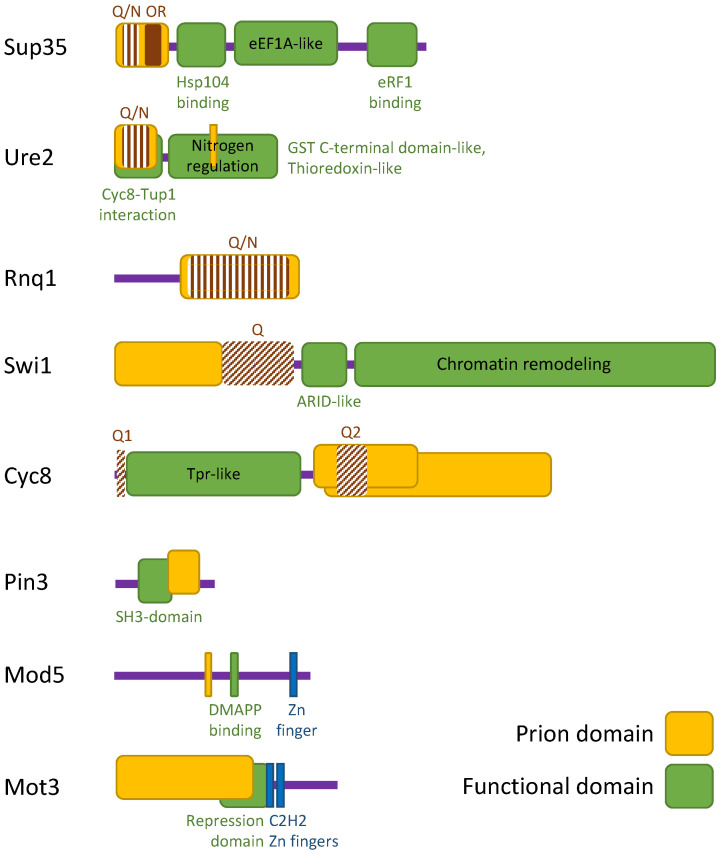
**Schematic representation of several yeast prion proteins. Sup35**: the N-terminal PrD contains a Q/N-rich region and oligopeptide repeats (OR), the middle part is involved in the interaction with Hsp104, the C-terminal part contains an eEF1A-like region with GTP-binding motifs and an interaction site with eRF1. **Ure2**: the PrD containing Q/N repeats is located at the N-terminus and overlaps with the site of interaction with the Cyc8-Tup1 corepressor. The C-part contains a functional region responsible for the regulation of nitrogen metabolism. This region shows high structural similarity to the β class glutathione S-transferases (GST) and contains a subdomain related to the thioredoxin-like superfamily. There is also an additional PrD within the C proximal part of Ure2. **Rnq1**: the Q/N-rich PrD is located on its C-terminal part. **Swi1**: the N-terminal region is N-rich and is required for prion formation. A middle Q-rich domain follows. The C terminal part contains the AT-rich interaction domain (ARID) and the region required for chromatin remodeling function. **Cyc8**: the PrD is located within the C-terminal part and shown in the scheme as two overlapping rounded squares since there are two options for the location of this domain. There are several Q-rich regions in Cyc8 marked in the scheme as Q1 and Q2. The N-terminus contains 10 tandem copies of the TPR (tetratricopeptide repeat) motif required for the interaction with repressor proteins. **Pin3**: the potential PrD located in its C-terminal part and overlaps with the functional SH3-domain. **Mod5**: the protein contains a short amyloid core closer to the center of the sequence. Behind it is a functional domain responsible for DMAPP (dimethylallyl diphosphate) binding and a zinc finger motif. **Mot3**: potential PrD is located in the N-terminal part. It overlaps with a functional repressor domain. There are two C2H2 zinc fingers at the C-terminus. The sizes of protein and functional blocks in the diagram are given taking into account their actual sizes. Information on functional domains is taken from the literature and the *Saccharomyces* Genome Database (https://www.yeastgenome.org/ accessed on 20 May 2023). See the text for more information.

**Figure 3 ijms-24-11651-f003:**
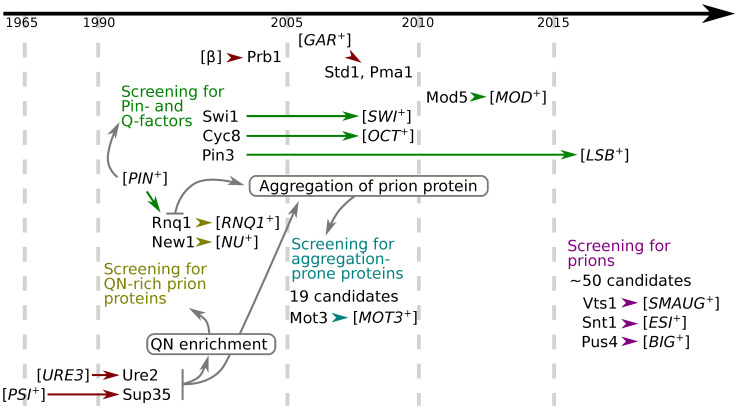
**Relationships between discoveries of yeast prions and corresponding proteins.** Colors correspond to the different approaches for prion identification and are the same as in Figure 1. Properties of prion proteins provided the basis for development of the new approaches are written in rectangles.

**Table 1 ijms-24-11651-t001:** *S. cerevisae* prions ^1^.

Prion	Protein Determinant	Protein Function	PrD Localization	PrD Properties
*1. Prions, discovered by their phenotype *				
[*PSI${}^{+}$*]	Sup35	Translation termination factor (eRF3)	N-term	Q/N rich
[*URE3*]	Ure2	Nitrogen catabolism regulation	N-term and secondary (C-term)	Q/N rich
[*GAR${}^{+}$*]	Pma1/Std1	Pma1—plasma membrane ATPase; Std1—regulator of cell response to glucose	ND ^3^	ND ^3^
[β]	Prb1	Vacuolar proteinase B, involved in protein degradation in the vacuole	ND ^3^	ND ^3^
*2. First prions, isolated from systematic screening for the identification of proteins with Q/N-rich sequences*				
[*NU${}^{+}$*]	New1	Translation termination or ribosome recycling	N-term	Q/N rich
[*RNQ${}^{+}$*] ^2^	Rnq1	ND ^3^	C-term	Q/N rich
*3. Prions, allowing the appearance of other prions*				
[*PIN${}^{+}$*] ^2^	Rnq1	ND ^3^	C-term	Q/N rich
[*SWI${}^{+}$*]	Swi1	Transcriptional regulator, component of the chromatin remodeling SWI/SNF complex	N-term	Q/N rich
[*OCT${}^{+}$*]	Cyc8	Transcriptional co-repressor (together with Tup1)	C-term	Q/N rich
[*LSB${}^{+}$*]	Pin3	Negative regulator of actin nucleation-promoting factor activity	C-term	Q/N rich
[*MOD${}^{+}$*]	Mod5	tRNA isopentenyltransferase	Middle part	ND ^3^
*4. Prions identified by combination of different approaches*				
[*MOT3${}^{+}$*]	Mot3	Transcription factor, modulates a variety of processes, including mating, carbon metabolism, stress response, and cell-wall remodeling	N-term	Q/N rich

^1^ See the text for references; ^2^ [*RNQ${}^{+}$*] and [*PIN${}^{+}$*] factors were discovered independently but actually represent the prion form of Rnq1; ^3^ ND —not determined.

## Data Availability

Data sharing is not applicable to this article.

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
