# Peer review of "How Big Is the Yeast Prion Universe?"

_ijms, 2023, doi:10.3390/ijms241411651_

Round 1
Reviewer 1 Report
The manuscript entitled “How big is the yeast prion universe?” offers an interesting review about yeast prions and approaches to identify them.
I do believe that the publication of this review will provide a useful guidance for researchers non-familiar (and also familiar) with the subject. However, there are some missing topics that I believe can be included in this review. For example:
1. What is a prion? The answer to this basic question is not given in the text! The phenomenological description of prions through their effects on living organisms as given in this review does not give an answer to this basic question.
2. The authors emphasized the genetics of prions, but they missed the biochemistry of prions. So, the mentioned literature is incomplete. Understanding the biochemistry of prions is essential. Hence, I do believe, for the sake of the readers, the author should include a bit insight on the current state of the art on prion’s biochemistry. For example:
i. Cobb NJ, Surewicz WK. Prion diseases and their biochemical mechanisms. Biochemistry. 2009;48: 2574–2585. doi:10.1021/bi900108v
ii. Zhao Y, Gong X, Jin S, Luo L. Research progress on prion protein structure and molecular mechanism of pathogenicity. Second International Conference on Biological Engineering and Medical Science (ICBioMed 2022). 2022. p. 42. doi:10.1117/12.2669372.
iii. Dennis EM, Garcia DM. Biochemical Principles in Prion-Based Inheritance. Epigenomes. 2022;6: 1–11. doi:10.3390/epigenomes6010004.
Author Response
The manuscript entitled “How big is the yeast prion universe?” offers an interesting review about yeast prions and approaches to identify them.
I do believe that the publication of this review will provide a useful guidance for researchers non-familiar (and also familiar) with the subject. However, there are some missing topics that I believe can be included in this review. For example:
- What is a prion? The answer to this basic question is not given in the text! The phenomenological description of prions through their effects on living organisms as given in this review does not give an answer to this basic question.
Response: We would like to thank the reviewer for the helpful suggestions and for a good opinion for the manuscript - we added the requested information in the beginning of the Introduction (lines 13-15):
“According to Stanley Prusiner, “prions are proteins that adopt alternative conformations, which are self-propagating and found in organisms ranging from yeast to humans” [3].”
- The authors emphasized the genetics of prions, but they missed the biochemistry of prions. So, the mentioned literature is incomplete. Understanding the biochemistry of prions is essential. Hence, I do believe, for the sake of the readers, the author should include a bit insight on the current state of the art on prion’s biochemistry. For example:
- Cobb NJ, Surewicz WK. Prion diseases and their biochemical mechanisms. Biochemistry. 2009;48: 2574–2585. doi:10.1021/bi900108v
- Zhao Y, Gong X, Jin S, Luo L. Research progress on prion protein structure and molecular mechanism of pathogenicity. Second International Conference on Biological Engineering and Medical Science (ICBioMed 2022). 2022. p. 42. doi:10.1117/12.2669372.
iii. Dennis EM, Garcia DM. Biochemical Principles in Prion-Based Inheritance. Epigenomes. 2022;6: 1–11. doi:10.3390/epigenomes6010004.
Response: We added the recommended discussion (lines 48-53):
“From the biochemical point of view prion conversion in many cases is based on conformational changes of corresponding proteins (for a review, see [16,21,22]). There is a single example when the protein digestion (prion [β]) is required for prion appearance [23]. Conformational changes associated with prionization often lead to the formation of amyloid aggregates, however most recent examples demonstrated that the prion aggregates may be non-amyloid ([24], for review, see [16]).”
Reviewer 2 Report
The manuscript ijms-2506033 is standard, in standard English, with minor linguistic and punctuational inaccuraciesI - need to be fixed
The authors are looking for new strategies for new yeast prions. The proteins formed in yeast have common characteristics, but also significant differences, such as: the functions of the protein; the ratio of Q and N; position relative to the protein sequence; the amount of protein in the cell, etc. The authors use the term "prion" and also consider the possibility that proteins can be divided into prions, quasiprions and prionoids.
They emphasize the fact that not only the intramolecular context — the amino acid composition — but also the intracellular one — the potential interactions of a protein in the cell — must be taken into account. I have no remarks about- introduction, уeast prions, problems and conclusion. The charts and tables used are accessible and well described. I am concerned by the fact that out of 106 references only 10 are from the last 3 years - need updating if possible.
Minor editing of English language required
Author Response
The manuscript ijms-2506033 is standard, in standard English, with minor linguistic and punctuational inaccuraciesI - need to be fixed.
The authors are looking for new strategies for new yeast prions. The proteins formed in yeast have common characteristics, but also significant differences, such as: the functions of the protein; the ratio of Q and N; position relative to the protein sequence; the amount of protein in the cell, etc. The authors use the term "prion" and also consider the possibility that proteins can be divided into prions, quasiprions and prionoids.
They emphasize the fact that not only the intramolecular context — the amino acid composition — but also the intracellular one — the potential interactions of a protein in the cell — must be taken into account. I have no remarks about- introduction, уeast prions, problems and conclusion. The charts and tables used are accessible and well described. I am concerned by the fact that out of 106 references only 10 are from the last 3 years - need updating if possible.
Response: We would like to thank the reviewer for the helpful suggestions and for a good opinion for the manuscript. The text was proofread and several mistakes were corrected.
Also, we added several references from the last 3 years ([11-19], lines 30-31). We tried to collect as much new information as possible about prions which were described some time ago, but this was not successful in all cases.